# Clinical and Immunological Factors Associated with the Progression of Lupus Nephritis in a Population from the Colombian Caribbean

**DOI:** 10.3390/biomedicines12092047

**Published:** 2024-09-09

**Authors:** María Vélez-Verbel, Gustavo Aroca-Martínez, David Vélez-Verbel, Alex Domínguez-Vargas, Manuela Vallejo-Patiño, Joanny Sarmiento-Gutierrez, Lorena Gomez-Escorcia, Carlos G. Musso, Henry J. González-Torres

**Affiliations:** 1Centro de Investigaciones en Ciencias de la Vida, Facultad de Ciencias de la Salud, Universidad Simón Bolívar, Barranquilla 080001, Colombia; maria.velez2@unisimon.edu.co (M.V.-V.); garoca1@hotmail.com (G.A.-M.); maria.vallejo1@unisimon.edu.co (M.V.-P.); joanny.sarmiento@unisimon.edu.co (J.S.-G.); carlos.musso@hospitalitaliano.org.ar (C.G.M.); 2Departamento de Nefrología, Clínica de la Costa, Barranquilla 080001, Colombia; lgomez72@unisimon.edu.co; 3Departamento de Medicina Interna, Clínica del Río, Magangué 132520, Colombia; davv1995@hotmail.com; 4División Ciencias de la Salud, Universidad del Norte, Barranquilla 080001, Colombia; dominguezaa@uninorte.edu.co; 5Data Analysis and Mining Department, D&P Consulting Service SAS, Barranquilla 080001, Colombia; 6Nephrology Department, Hospital Italiano de Buenos Aires, Buenos Aires C1000, Argentina; 7Doctorado en Ciencias Biomédicas, Universidad del Valle, Cali 760001, Colombia

**Keywords:** lupus nephritis, immunological factors, disease progression

## Abstract

Lupus nephritis represents a significant immune-mediated glomerulonephritis, constituting the most important organ involvement induced by systemic lupus erythematosus (SLE), with variable epidemiology and clinical presentation among populations. Objective: to identify clinical and immunological factors associated with the progression of lupus nephritis in a population from the Colombian Caribbean. Methods: we evaluated 401 patients diagnosed with SLE and lupus nephritis, treated at a reference center in the Colombian Caribbean, gathering data recorded in medical records. Results: A proportion of 87% were female, with a median age of 42 years. Most patients presented with proliferative classes (90%), with class IV being the most common (70%). A proportion of 52% of patients did not respond to treatment, which is described as the lack of complete or partial response, while 28% had a complete response. A significant decrease in hemoglobin, glomerular filtration rate, and proteinuria was identified by the third follow-up (*p* < 0.001), along with an increase in creatinine, urea, and hematuria (*p* < 0.001). Patients with initial proteinuria > 2 g/day were found to be 27 times more likely to be non-responders (*p* < 0.001). Mortality was associated with the presence of serum creatinine >1.5 mg/dL (*p* = 0.01) (OR: 1.61 CI 95% 0.75–3.75) and thrombocytopenia (*p* = 0.01) (OR: 0.36; CI 95% 0.12–0.81). Conclusions: identifying factors of progression, non-response, and mortality provides an opportunity for more targeted and personalized intervention, thereby improving care and outcomes for patients with lupus nephritis.

## 1. Introduction

Lupus nephritis (LN) is the most frequently affected organ by systemic lupus erythematosus (SLE), and it is associated with disease severity, leading to higher morbidity and mortality [1]. Its presence has been described in 40–70% of patients with SLE within five years of diagnosis, characterized by abnormalities in renal function tests due to glomerular, tubulointerstitial, and vascular lesions. Additionally, progression to end-stage renal disease has been reported in 4.3–20% of cases [2,3].

Epidemiologically, the prevalence of LN varies according to factors such as race and ethnicity. European populations have a lower risk of developing LN, whereas Asians and Hispanics exhibit renal involvement in 82% and 50% of cases, respectively. The prevalence in the Afro-descendant population is 50.7%, compared to 39.5% in whites. Furthermore, there is an association between socio-economic status (economic instability, poverty, unemployment, food insecurity, stress, racial segregation, discrimination, etc.) and the development of LN in 14.5% of cases, genetic factors (genes associated with programmed cell death, immune complex clearance, and adaptive immunity) in 36.8%, and a combination of both in 12.2% [4,5,6,7]. In Colombia, the prevalence of LN in adult patients diagnosed with SLE has been described as 50–55%, corresponding to approximately 54.9% in the department of Atlántico (Colombia) [8].

In all patients diagnosed with SLE, the concomitant presence of LN should be suspected. Routine evaluation should include active urinary sediment and proteinuria, as well as the positivity of anti-double-stranded DNA antibodies (anti-dsDNA). However, definitive diagnosis is confirmed by renal biopsy, which also allows for histological classification and targeted therapy [9,10].

Despite new therapies developed to improve the balance between efficacy and toxicity, unmet needs persist in LN. A considerable mortality rate is associated with the disease, with a survival rate of 72% in patients with focal proliferative glomerulonephritis and 67% for diffuse glomerulonephritis at five years, regardless of treatment [11,12,13].

Considering the above, it is evident that LN remains the most severe complication of SLE, responsible for high morbidity, progression to end-stage chronic kidney disease, and mortality. Currently, there is limited evidence to establish the risk of disease progression, highlighting the need for clinical and/or immunological variables to aid in establishing prognosis.

## 2. Materials and Methods

### 2.1. Subjects

This study enrolled patients aged 18 to 80 years diagnosed with systemic lupus erythematosus (SLE) and LN identified through renal biopsy in all patients. Recruitment was conducted at the nephrology department of a tertiary hospital located in Barranquilla, Colombia, serving as a reference center for the entire Colombian Caribbean region. All patients met four or more of the criteria for SLE proposed by the EULAR/ACR [1]. Patients under 18 years of age were excluded, as well as those in whom the diagnostic criteria were not accurately met. Also, patients with advanced damage not susceptible to immunosuppression treatment (class VI) were not included in the present study.

### 2.2. Variables

Demographic data (age, sex) and clinical information (disease duration, LN class, disease activity, comorbidities, medication) were collected upon consent. Disease activity was assessed using the Systemic Lupus Erythematosus Disease Activity Index 2000 (SLEDAI-2K) [2]. Patients who were not responsible for managing their medication or unable to provide informed consent were excluded.

### 2.3. Statistical Methods

Normality of data was assessed using the Kolmogorov–Smirnov test. Quantitative variables are presented as medians with Q_1_-Q_3_ ranges. Absolute and relative frequencies are used to describe categorical variables.

The Wilcoxon rank-sum test was used to evaluate median differences between patients by sex, histological type (proliferative vs. non-proliferative class), and survival status (survivors vs. deceased). The Kruskal–Wallis test was employed to compare median differences across three clinical follow-up time points. Fisher’s exact test and the Chi^2^ test were used to analyze categorical variables.

Multivariate logistic regression analysis, adjusted with a backward methodology, was performed to assess potential clinical, immunological, and histopathological factors associated with treatment non-response and mortality. Odds Ratios, 95% Confidence Intervals, and *p*-values are reported. A significance level of *p* < 0.05 was considered statistically significant. Statistical analysis was conducted using R-CRAN version 4.3.2.

## 3. Results

### 3.1. General Characteristics

A total of 401 patients diagnosed with LN confirmed by renal biopsy participated in this study, of whom 87% were female. The median age was 42 years (Q_1_-Q_3_: 11–93). Regarding histological type, many patients had proliferative classes (90%), distributed as follows: class IV (70%) and class III (20%).

Regarding treatment response, most patients (*n* = 210, 52%) were non-responders, while 114 (28%) patients showed a complete response. The most frequently used immunosuppressant was Mycophenolate Mofetil (MMF) (*n* = 281, 70%), used in combination with Cyclophosphamide (CTX) in 26 cases (6.5%), and all the patients were receiving glucocorticoids. In this study, 61 patients (15%) died (Table 1).

### 3.2. Clinical, Immunological, and Histopathological Profile by Sex

In this study, we compared clinical, immunological, and histopathological characteristics in patients with LN by sex (Table 2). No significant differences were observed in age, complete blood count, immunological parameters, histopathological findings, or survival (all *p* > 0.05).

On the other hand, we observed statistically significant differences in 24 h proteinuria, which was higher in males (1.7 g/day, Q_1_-Q_3_: 0.1–9.1) compared to females (1.3 g/day, Q_1_-Q_3_: 0.1–13.5) (*p* = 0.004). Additionally, a significant difference in estimated glomerular filtration rate (eGFR) was noted, with females exhibiting a lower eGFR (54 mL/min/m^2^, Q_1_-Q_3_: 6–168) compared to males (77 mL/min/m^2^, Q_1_-Q_3_: 30–214) (*p* < 0.001) (Figure 1).

In terms of treatment response, a significant association was observed (*p* = 0.03), with higher rates of complete response in women (30%) compared to men (19%), and a greater proportion of non-responders among males (69% vs. 50%) (Figure 2).

### 3.3. Clinical, Immunological, and Histopathological Profile by Histological Type

In this study, patients with LN were classified according to histological type into proliferative (class III-IV) and non-proliferative (I-II-V) classes. The clinical, immunological, and histopathological profiles were compared between both groups.

We observed that patients with non-proliferative classes were significantly older compared to the proliferative class group (46 years, Q_1_-Q_3_: 27–83 vs. 41 years, Q_1_-Q_3_: 19–93) (*p* = 0.02). Overall, the groups were similar in terms of hemogram parameters, renal function, immunological indices, and outcomes (treatment response and survival) (all *p* > 0.05) (Table 3).

### 3.4. Evolution of Clinical and Immunological Parameters

To evaluate the medical follow-up of patients with LN, three controls were established in the measurement of hemogram, renal function, and immunological parameters. The first control was taken before the start of treatment, the second in the middle of the induction treatment, and the third control at the end of it, and death was assessed during this time (Table 4).

A significant decrease was observed in the median of hemoglobin, glomerular filtration rate, and proteinuria towards the third control (all *p* < 0.001). In contrast, a significant increase was observed in the median of serum creatinine and urea, and in the proportion of patients with hematuria towards the third control (*p* < 0.001). No significant differences were observed in platelet levels, anti-ds-DNA titration, or in the proportion of patients with C3/C4 hypocomplementemia (all *p* > 0.05) (Figure 3).

### 3.5. Clinical, Immunological, and Histopathological Characteristics According to Survival

Patients were analyzed according to survival (survivors, *n* = 340; deceased, *n* = 61), considering demographic, clinical, immunological, and histopathological parameters. No significant differences were observed in demographic or histopathological parameters, nor in the proportions of patients by treatment response (all *p* > 0.05) (Table 5).

A significant difference was observed in platelet count, being higher in deceased patients (259 μL, IQR: 91–443) compared to survivors (222 μL, Q_1_-Q_3_: 76–450) (*p* = 0.01). Regarding renal function, a higher median of serum creatinine was observed in deceased patients (1.4 mg/dL, Q_1_-Q_3_: 0.50–8.10) compared to survivors (1.2 mg/dL, Q_1_-Q_3_: 0.50–12) (*p* = 0.01). Consequently, lower glomerular filtration rates were evidenced in deceased patients (48 mL/min/m^2^, Q_1_-Q_3_: 6–151) compared to survivors (58 mL/min/m^2^, Q_1_-Q_3_: 8–214) (*p* = 0.01). Regarding immunological parameters, the prevalence of anti-dsDNA positivity was significantly higher in deceased patients than in survivors (69% vs. 51%, *p* = 0.03) (Figure 4).

### 3.6. Predictors of Non-Response to Treatment

Possible factors related to non-response to treatment in LN patients were evaluated through multivariate logistic regression analysis (Table 6). In the adjusted model, it was evident that male patients exhibited a 1.9-fold increased likelihood of non-response to treatment (OR: 1.92; 95% CI: 1.62–4.59, *p* = 0.04) in comparison to females. Patients with initial proteinuria > 2 g/day were 27 times more likely to not respond (OR: 27.3, 95% CI: 15.9–54.1; *p* < 0.001). Similarly, patients with C3 hypocomplementemia had a higher risk of not responding to treatment (OR: 1.8; 95% CI: 1.27–3.89, *p* = 0.02) compared to those patients with normal Complement C3 values (Table 6).

### 3.7. Predictors of Mortality in LN Patients

In this study, a multivariate logistic regression analysis was conducted to identify possible factors associated with mortality in LN patients (Table 7). In the adjusted model, we observed that patients with serum creatinine >1.5 mg/dL were twice more likely to die than those with levels <1.5 mg/dL (OR: 2.08; 95% CI 1.16–3.9, *p* = 0.01). Furthermore, patients with proliferative histological classes (III-IV) were 1.8 times more likely to die compared to non-proliferative classes (I-II-V) (OR: 1.82, 95% CI: 1.6–8.6, *p* = 0.03). In contrast, patients with platelet count >150 μL had a 66% reduction in the risk of dying (OR: 0.34, 95% CI: 0.12–0.75, *p* = 0.01).

## 4. Discussion

Lupus nephritis (LN) is a severe and prevalent manifestation of systemic lupus erythematosus (SLE) [3]. This retrospective study encompasses one of the largest sample sizes in Colombia, offering significant insights into this condition. Within the findings of this study, 87% were females with a median age of 42 years, consistent with global epidemiological reports, showing predominantly female involvement under the age of 55 [4]. Most patients presented with proliferative classes (90%), with Class IV being the most common (70%). The high prevalence of these findings is noteworthy, with varied reports ranging from 68 to 81% for proliferative classes across different populations [5,6].

Regarding treatment, Mycophenolate Mofetil was the most used immunosuppressant in 70% of cases as a standard dose of 2–3 g/day. The treatment response of LN was evaluated as a complete response, partial response, or non-response; in this study, 52% of evaluated patients showed no response to treatment, with only 28% achieving total response, consistent with other reports where the total response was around 20.2%, possibly reflecting the severity of the disease as it was reported by Izmirli et al. in which only few patients had, at week 12, a complete response sustained through the entire year of the study, and only 26% attained a partial or complete response at both week 26 and week 52 [7]. In contrast, other cohorts such as Park et al. have described complete, partial, and no response rates at 6 and 12 months of 52.0%, 26.7%, and 21.3%, and 50.7%, 24.0%, and 25.3%, respectively, while Choi et al. found responses of 43.8%, 25.0%, and 31.2%, and 56.2%, 18.8%, and 25.0%, respectively, at 6 and 12 months [9,10].

In other studies, longer follow-ups were carried out over time, as described by Du et al. A total of 48.2% and 75.0% of patients achieved a complete response after standard treatment for 6 months and 2 years, drawing attention to the more aggressive behavior of the disease in our population expressed as an evolution over medical follow-up with significant decreases in hemoglobin, glomerular filtration rate, and proteinuria by the third follow-up, with increases in serum creatinine, urea, and hematuria. These changes highlight the disease dynamics and suggest the importance of continuous monitoring to adapt treatment and prevent progression and natural evolution, given the tendency towards renal functional deterioration and progression to end-stage chronic kidney disease [11,12].

No significant differences were observed in clinical, immunological, or histopathological parameters based on sex, except for higher proteinuria but higher glomerular filtration rate in men. Furthermore, a higher proportion of non-response to treatment was observed in males correlating with male sex that was a determinant for the early development and progression of LN and chronic kidney disease, subsequently resulting in a lesser treatment response compared to females. This underscores the importance of sex as a significant variable in evaluating clinical presentation and treatment response [13]. This agrees with what was described by Rafifi et al. and Resende et al. in whose cohorts male gender presented a worse evolution of LN [14,15].

Histopathological classification revealed age differences, with patients in non-proliferative classes being significantly older. Although no significant differences were observed in paraclinical findings, treatment response, or survival, consistent with Ichinose et al., who reported that age at onset of LN did not correlate with renal response at 12 months, we considered that a more detailed evaluation of renal histology in LN is warranted [16,17].

Regarding predictors of treatment non-response, the presence of proteinuria exceeding 2 g/day prior to treatment was associated with a 27 times higher odds of treatment non-response, similar to a report by Luis et al. [18], in contrast with what was described by Delfino J et al. and Bobirca et al. who found no difference in response in relation to proteinuria levels [19,20]. Additionally, hypocomplementemia C3 in this study was significantly related to treatment non-response, describing it as a strong predictor of progression to end-stage chronic kidney disease [21], differing from what was described in the population of Miranda-Hernandez et al. who demonstrated that low C3 was associated with good response to treatment [22], while for Dall’era et al., C4 levels at the beginning of treatment are decisive for the response [23].

On the other hand, no significant differences were found in demographic, histopathological, or treatment response parameters in relation to mortality; meanwhile, greater renal involvement, indicated by serum creatinine > 1.5 mg/dL and proliferative histological classes, proved to be important predictors of mortality possibly in relation to the greater cardiovascular risk conferred by it [23]. Other studies have shown that acute kidney injury, a lack of remission at one year, and non-adherence are associated with poor prognosis in terms of mortality. Variables such as male sex have also been found to be important predictors of mortality, indicating that findings vary across different populations and the comprehension of these findings can guide diverse and early therapeutic strategies to improve outcomes in at-risk populations [24,25,26].

This study represents the real-world standard of care, including patients with established damage caused by LES and its complications. Several studies have consistently shown that patients from Hispanic backgrounds develop LN early and have more aggressive disease, which can explain the large differences found in this population when compared to others [7,27].

## 5. Conclusions

LN in the Colombian Caribbean population, as demonstrated by the population included in the present study, shows its complexity and variability in terms of clinical presentation and therapeutic response. Special consideration of sex, histopathology, and the dynamic clinical evolution of various parameters is essential for effective management. Also, this study highlights the importance of understanding sex differences in clinical presentation and treatment response. Identifying predictive factors such as proteinuria, creatinine levels, proliferative class, and hypocomplementemia is crucial for early and personalized interventions. This approach can improve care and outcomes for patients with LN, as morbidity and mortality rates remain high despite advances in novel treatments.

## Figures and Tables

**Figure 1 biomedicines-12-02047-f001:**
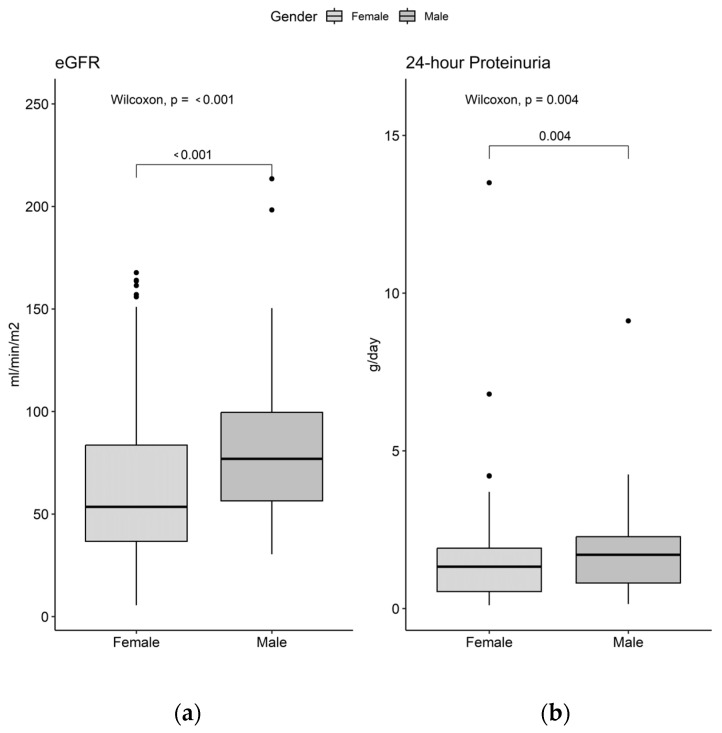
Glomerular filtration rate (**a**) and 24 h proteinuria (**b**) in patients with LN by sex.

**Figure 2 biomedicines-12-02047-f002:**
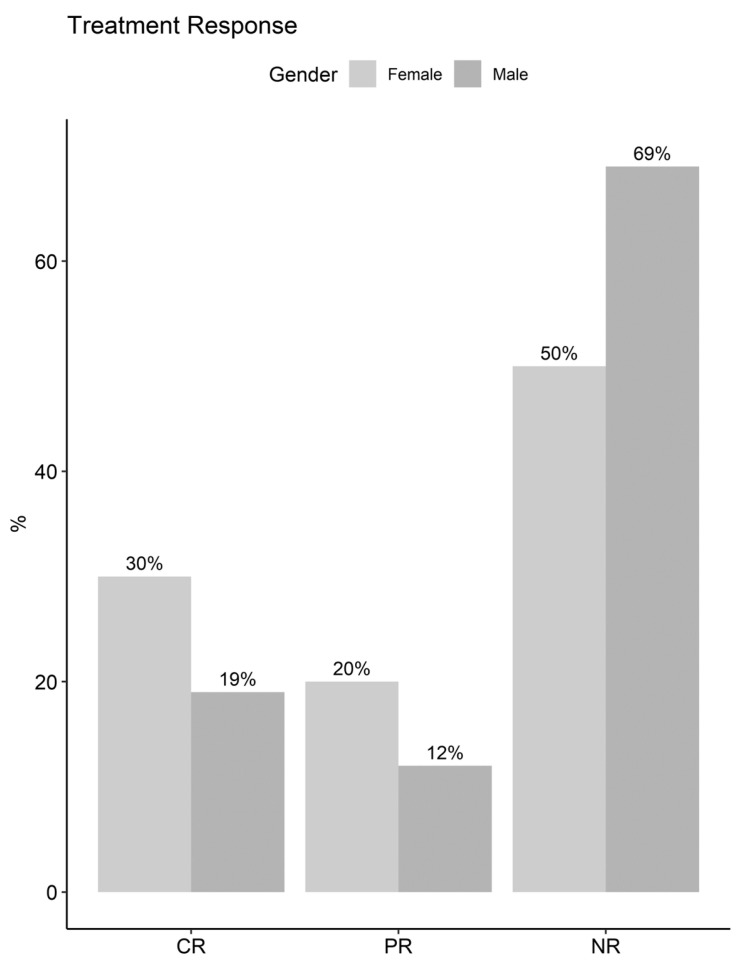
Distribution of treatment response by sex. CR: complete response, PR: partial response, NR: no response.

**Figure 3 biomedicines-12-02047-f003:**
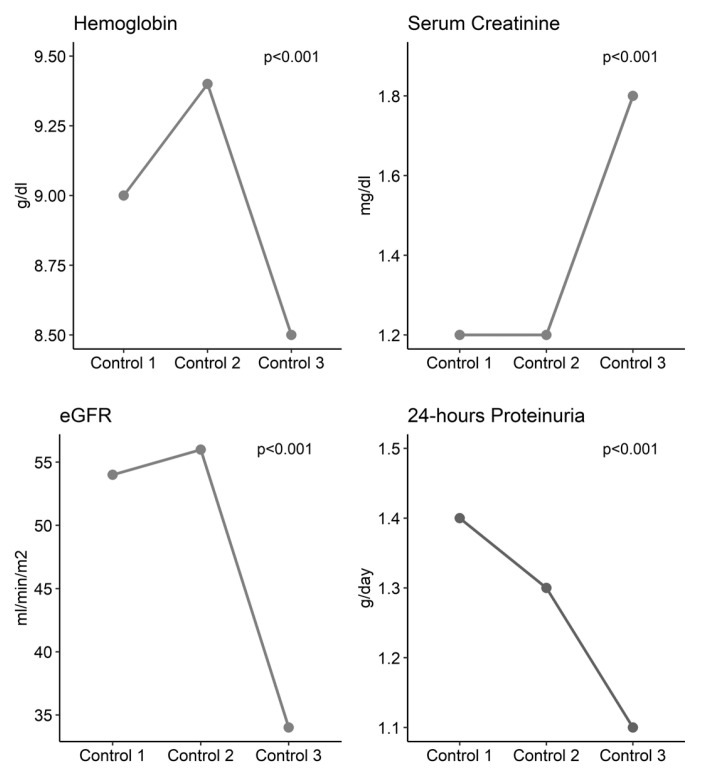
Three-control follow-up of median hemoglobin, serum creatinine, glomerular filtration rate, and 24 h proteinuria.

**Figure 4 biomedicines-12-02047-f004:**
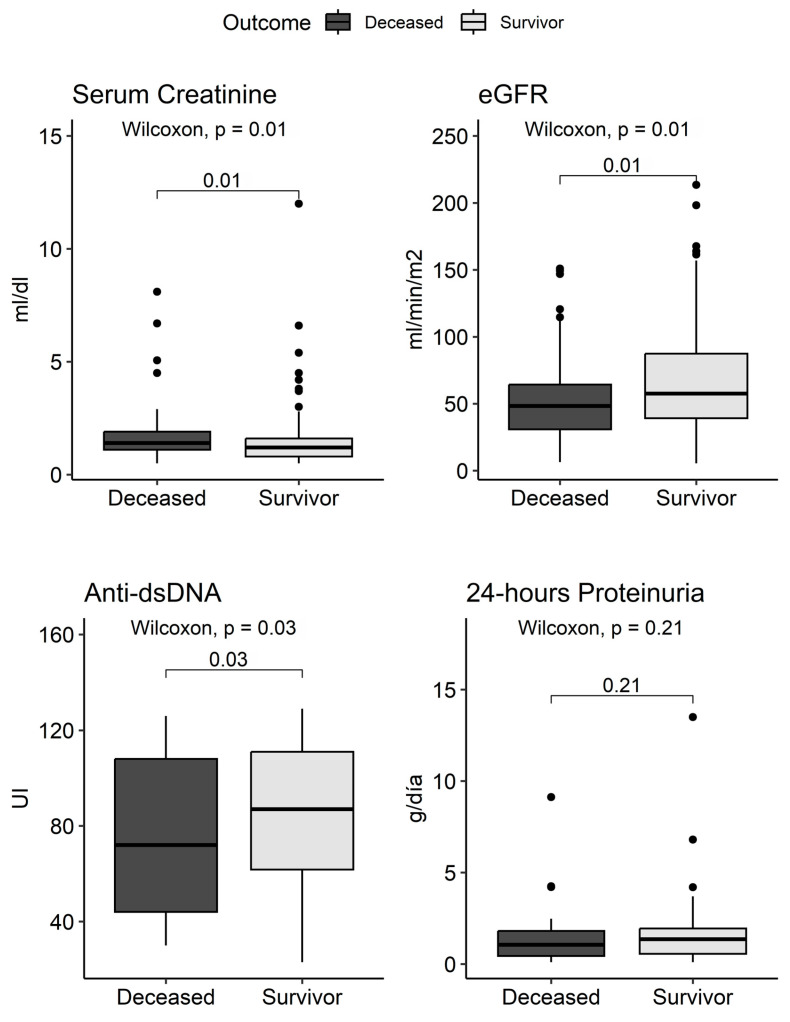
Serum creatinine, glomerular filtration rate, anti-ds-DNA levels, and 24 h proteinuria according to survival.

**Table 1 biomedicines-12-02047-t001:** General characteristics in patients with LN.

Characteristic	*n* = 401 ^1^
Age	42 (19, 93)
Sex	
Female	349 (87%)
Male	52 (13%)
Histological Class	
I	2 (0.5%)
II	26 (6.5%)
III	79 (20%)
IV	282 (70%)
V	12 (3.0%)
VI	0 (0.0%)
Histological type	
Non-proliferative	40 (10.0%)
Proliferative	361 (90%)
Treatment Response	
CR	114 (28%)
PR	77 (19%)
NR	210 (52%)
Immunosuppressant	
MMF + CTX	26 (6.5%)
CTX	53 (13%)
MMF	281 (70%)
Survival	
Deceased	61 (15%)
Survivor	340 (85%)

CR: complete response; PR: partial response; NR: no response; MMF: Mycophenolate Mofetil; CTX: Cyclophosphamide; ^1^ median (range); *n* (%).

**Table 2 biomedicines-12-02047-t002:** Clinical, immunological, and histopathological profile of patients with LN by sex.

Parameter		Female (*n* = 349) ^1^	Male, (*n* = 52) ^1^	*p*-Value
Age		41 (19, 93)	43 (20, 75)	0.71 ^2^
Hemogram	Hb (g/dL)	9.4 (5.2, 14.5)	9.6 (5.2, 14.2)	0.62 ^2^
	Plt (10^3^/μL)	225 (76, 450)	250 (76, 400)	0.41 ^2^
	Leu (mm^3^ × 10^3^)	7.9 (2.3, 14.4)	7.5 (2.5, 12.6)	0.43 ^2^
Renal Function	sCr (mg/dL)	1.2 (0.5, 15)	1.1 (0.5, 2.4)	0.63 ^2^
	eGFR (mL/min/m^2^)	54 (6, 168)	77 (30, 214)	<0.001 ^2^
	Urea (mg/dL)	60 (14, 118)	61 (16, 118)	0.82 ^2^
	Proteinuria (g/day)	1.3 (0.1, 13.5)	1.7 (0.1, 9.1)	0.004 ^2^
	Hematuria	186 (53%)	23 (44%)	0.21 ^3^
Immunological	Hypocomplementemia C3	257 (74%)	35 (67%)	0.32 ^3^
	Hypocomplementemia C4	73 (21%)	11 (21%)	>0.9 ^3^
	Anti ds DNA (% positive)	122 (35%)	16 (31%)	0.61 ^3^
Histopathology	Histological Type			0.72 ^3^
	Non-proliferative (I,II,V)	34 (9.7%)	6 (12%)	
	Proliferative (III,IV)	315 (90%)	46 (88%)	
	Activity Index	6 (1, 15)	5.5 (1, 13)	0.54 ^2^
	Chronicity Index	4 (0, 9)	4.5 (0, 9)	0.91 ^2^
Outcome	Treatment Response			0.03 ^3^
	CR	104 (30%)	10 (19%)	
	NR	174 (50%)	36 (69%)	
	PR	71 (20%)	6 (12%)	
	Survival			0.72 ^3^
	Deceased	52 (15%)	9 (17%)	
	Survivor	297 (85%)	43 (83%)	

Hb: hemoglobin; Plt: platelets; Leu: Leukocytes; sCr: serum creatinine; eGFR: estimated glomerular filtration rate; C3: Complement C3; C4: Complement C4; CR: complete response, NR: no response; PR: partial response; ^1^ median (range); *n* (%); ^2^ Wilcoxon rank sum test; ^3^ Pearson’s Chi-squared test.

**Table 3 biomedicines-12-02047-t003:** Clinical, immunological, and histopathological characteristics of patients with LN by histological type.

Characteristic		Non-Proliferative (I-II-V) (*n* = 40) ^1^	Proliferative (III-IV) (*n* = 361) ^1^	*p*-Value
Demographics	Age	46 (27, 83)	41 (19, 93)	0.02 ^2^
	Sex			0.71 ^3^
	Female	34 (85%)	315 (87%)	
	Male	6 (15%)	46 (13%)	
Hemogram	Hemoglobin (g/dL)	9.9 (5.4, 14.3)	9.4 (5.2, 14.5)	0.13 ^2^
	Platelets (10^3^/μL)	241 (84, 443)	228 (76, 450)	0.42 ^2^
	Leu (mm^3^ × 10^3^)	7.8 (3.7, 14.2)	8.0 (2.3, 14.4)	0.93 ^2^
Renal Function	Serum creatinine (mg/dL)	1.1 (0.5, 4.5)	1.2 (0.5, 12)	0.41 ^2^
	eGFR (mL/min/m^2^)	63 (12, 198)	55 (6, 214)	0.62 ^2^
	Urea (mg/dL)	58 (14, 118)	61 (15, 118)	0.41 ^2^
	Proteinuria (g/day)	1.23 (0.14, 2.48)	1.36 (0.1, 13.5)	0.53^2^
	Hematuria	19 (48%)	190 (53%)	0.52 ^3^
Immunological	Hypocomplementemia C3	30 (75%)	262 (73%)	0.71 ^3^
	Hypocomplementemia C4	7 (18%)	77 (21%)	0.62 ^3^
	Anti ds DNA (% positive)	14 (35%)	124 (34%)	0.92 ^3^
Histopathology	Activity Index	5.0 (1.0, 13.0)	6.0 (1.0, 15.0)	0.31 ^2^
	Chronicity Index	5.00 (0.00, 9.00)	4.00 (0.00, 9.00)	0.14 ^2^
Outcome	Treatment Response			0.44 ^3^
	Complete Response (CR)	14 (35%)	100 (28%)	
	No Response (NR)	21 (53%)	189 (52%)	
	Partial Response (PR)	5 (13%)	72 (20%)	
	Survival			0.23 ^3^
	Deceased	3 (7.5%)	58 (16%)	
	Survivor	37 (93%)	303 (84%)	

Leu: Leukocytes; eGFR: estimated glomerular filtration rate; C3: Complement C3; C4: Complement C4; CR: complete response, NR: no response; PR: partial response; ^1^ median (range); *n* (%); ^2^ Wilcoxon rank sum test; ^3^ Pearson’s Chi-squared test

**Table 4 biomedicines-12-02047-t004:** Evolution of clinical and immunological parameters over medical follow-up in patients with LN.

Characteristic		First Control	Second Control	Third Control	*p*-Value
Hemogram	Hb (g/dL)	9 (4.9, 14.5)	9.4 (5.2, 14.5)	8.5 (5.2, 12)	<0.001 ^1^
	Plt (10^3^/μL)	226 (328, 723)	230 (76, 450)	224 (76, 349)	0.84 ^1^
	Leu (mm^3^ × 10^3^)	7.9 (5.0, 71)	7.9 (2.3, 14.4)	8.3 (2.5, 15.0)	0.31 ^1^
Renal Function	sCr (mg/dL)	1.20 (0.40, 8.30)	1.2 (0.5, 12)	1.8 (0.5, 4.5)	<0.001 ^1^
	eGFR (mL/min/m^2^)	54 (6, 223)	56 (6, 214)	34 (12, 181)	<0.001 ^1^
	Urea (mg/dL)	43 (10, 223)	60 (14, 118)	82 (14, 149)	<0.001 ^1^
	Proteinuria (g/day)	1.42 (0.13, 12.10)	1.35 (0.1, 13.5)	1.1 (0.1, 7.5)	<0.001 ^1^
	Hematuria	211 (53%)	209 (52%)	241 (61%)	<0.001 ^2^
Immunological	Hipocomplementemia C3	250 (62%)	292 (73%)	310 (77%)	0.19 ^2^
	Hipocomplementemia C4	81 (20%)	84 (21%)	110 (27%)	0.63 ^2^
	Anti ds DNA (% positive)	138 (34%)	149 (37%)	126 (32%)	0.23 ^2^

Hb: hemoglobin; Plt: platelets; Leu: Leukocytes; sCr: serum creatinine; eGFR: estimated glomerular filtration rate; C3: Complement C3; C4: Complement C4; ^1^ Kruskal–Wallis rank sum test; ^2^ Pearson’s Chi-squared test.

**Table 5 biomedicines-12-02047-t005:** Clinical, immunological, and histopathological features of LN patients according to survival.

Characteristic		Deceased (*n*= 61) ^1^	Survivor (*n* = 340) ^1^	*p*-Value
Demographical	Age	44 (24, 69)	41 (19, 93)	0.08 ^2^
	Sex			0.71 ^3^
	Female	52 (85%)	297 (87%)	
	Male	9 (15%)	43 (13%)	
Hemogram	Hb (g/dL)	9.4 (5.2, 13.3)	9.45 (5.2, 14.5)	>0.9 ^2^
	Plt (10^3^/μL)	259 (91, 443)	222 (76, 450)	0.01 ^2^
	Leu (mm^3^ × 10^3^)	8.09 (3.1,14.3)	7.9 (2.3, 14.4)	>0.9 ^2^
Renal Function	sCrt (mg/dL)	1.40 (0.50, 8.10)	1.20 (0.50, 12)	0.01 ^2^
	eGFR (mL/min/m^2^)	48 (6, 151)	58 (8, 214)	0.01 ^2^
	Urea (mg/dL)	62 (14, 118)	60 (15, 118)	0.82 ^2^
	Proteinuria (g/day)	1.06 (0.1, 9.1)	1.36 (0.1, 13.5)	0.23 ^2^
	Hematuria	34 (56%)	175 (51%)	0.51 ^3^
Immunological	C3 hypocomplementemia	48 (79%)	244 (72%)	0.34 ^3^
	C4 hypocomplementemia	17 (28%)	67 (20%)	0.15 ^3^
	Anti ds DNA (% positive)	42 (69%)	173 (51%)	0.01 ^3^
Histopathological	Histological Type			0.22 ^3^
	Non-proliferative	3 (4.9%)	37 (11%)	
	Proliferative	58 (95%)	303 (89%)	
	Activity Index	7 (1, 15)	6 (1, 14)	>0.9 ^2^
	Chronicity Index	5 (0, 9)	4 (0, 9)	0.61 ^2^
Outcome	Response to Treatment			0.12 ^3^
	CR	24 (39%)	90 (26%)	
	NR	27 (44%)	183 (54%)	
	PR	10 (16%)	67 (20%)	

Hb: hemoglobin; Plt: platelets; Leu: Leukocyte; sCrt: serum creatinine; eGFR: estimated glomerular filtration rate; C3: Complement C3; C4: Complement C4; CR: complete response, NR: no response; PR: partial response; ^1^ Medium (Range); *n* (%); ^2^ Wilcoxon rank sum test; ^3^ Pearson’s Chi-squared test.

**Table 6 biomedicines-12-02047-t006:** Multivariate logistic regression analysis of predictors for non-response in patients with NL.

Variable	Multivariate			Adjusted		
	OR ^1^	95% CI ^2^	*p*-Value	OR ^1^	95% CI ^2^	*p*-Value
Age > 30						
Yes	0.73	0.40, 1.27	0.31	0.76	0.43, 1.32	0.31
Sex						
Female	—	—		—	—	
Male	2.14	0.96, 5.36	0.07	1.92	1.62, 4.59	0.04
Hb < 10 g/dL						
No	—	—		—	—	
Yes	0.68	0.39, 1.17	0.22	0.65	0.38, 1.10	0.11
Plt > 150 10^3^/μL						
Yes	0.75	0.39, 1.42	0.41			
PRT > 2 g/day						
Yes	28.4	16.4, 57.9	<0.001	27.3	15.9, 54.1	<0.001
sCRT > 1.5 mg/dL						
No	—	—				
Yes	0.96	0.46, 1.98	>0.9			
eGFR < 60 mL/min/m^2^						
Yes	1.45	0.72, 3.15	0.32			
Hematuria						
Yes	0.96	0.56, 1.64	0.91			
Hypocomplementemia C3						
Yes	1.51	1.17, 3.91	0.03	1.8	1.27, 3.89	0.02
Hypocomplementemia C4						
Yes	0.68	0.34, 1.30	0.23			
Anti ds DNA Positive						
Yes	1.46	0.70, 3.78	0.71			
Histologic type						
Non-proliferative	—	—				
Proliferative	0.75	0.31, 1.77	0.52			
Activity Index > 7						
Yes	1.03	0.60, 1.77	>0.9			
Immunosuppressor Treatment						
CTX	—	—				
MMF	1.25	0.56, 2.97	0.64			

Hb: hemoglobin; Plt: platelets; PRT: Proteinuria; sCRT: Serum creatinine; eGFR: estimated glomerular filtration rate; CTX: Cyclophosphamide; MMF: Mycophenolate Mofetil; ^1^ OR: Odds Ratio; ^2^ CI: Confidence Interval.

**Table 7 biomedicines-12-02047-t007:** Multivariate logistic regression analysis of predictive factors for mortality in patients with LN.

Parameter	Multivariate			Adjusted		
	OR ^1^	95% CI ^2^	*p*-Value	OR ^1^	95% CI ^2^	*p*-Value
Age > 35						
Yes	1.55	0.81, 3.25	0.2	1.62	0.87, 3.34	0.14
Sex						
Females	—	—				
Male	1.31	0.48, 3.43	0.6			
Plt > 150 μL						
Yes	0.36	0.12, 0.81	0.024	0.34	0.12, 0.75	0.015
PRT > 1 g/day						
Yes	1.41	0.61, 3.74	0.4			
sCRT > 1.5 mg/dL						
Yes	1.61	0.75, 3.75	0.2	2.08	1.16, 3.93	0.016
eGFR < 60 mL/min/m^2^						
Yes	1.46	0.59, 3.60	0.4			
Hematuria						
Yes	1.02	0.54, 1.94	>0.9			
Hypocomplementemia C3						
Yes	1.07	0.52, 2.29	0.9	1.17	0.60, 2.47	0.6
Hypocomplementemia C4						
Yes	1.69	0.81, 3.54	0.15			
Anti ds DNA Positive						
Yes	2.6	0.45, 8.75	0.07			
Histological Type						
Non-proliferative	—	—		—	—	
Proliferative	1.65	0.55, 7.91	0.4	1.82	1.63, 8.68	0.03
Activity Index > 6						
Yes	1.36	0.73, 2.61	0.3			
Response to Treatment						
No	—	—				
Yes	1.71	0.72, 4.60	0.2			
Immunosuppressor						
CTX	—	—		—	—	
MMF	1.99	0.82, 6.59	0.2	2.05	0.86, 6.65	0.14

Plt: platelets; PRT: Proteinuria; sCRT: Serum creatinine; eGFR: estimated glomerular filtration rate; CTX: Cyclophosphamide; MMF: Mycophenolate Mofetil; ^1^ CI: Confidence Interval. ^2^ OR: Odds Ratio.

## Data Availability

All data underlying the results are available as part of this article.

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
