# Peer review of "Clinical and Immunological Factors Associated with the Progression of Lupus Nephritis in a Population from the Colombian Caribbean"

_biomedicines, 2024, doi:10.3390/biomedicines12092047_

Round 1
Reviewer 1 Report
Comments and Suggestions for Authors
This is an interesting article by María Vélez-Verbel et al.
I would like to address a small number of suggestions to you.
General recommendations
Please correct all references according to journal stile.
References should be described as follows:
Journal Articles:
1. Author 1; Author 2. Title of the article. Abbreviated Journal Name Year; Volume: page range.
Page 1, line 39
Please write abbreviation for Lupus nephritis.
Material and methods
Page 2
Exclusion criteria must be written in section "Subjects"
Please write dentally methods for all immunological and biochemical parameters that were used in this study.
Is it necessary to write p value in the figure 1 in the duplicate?
For which parameters p value is 0.03?
Page 6, figure 2
For which parameters p value is 0.03?
It is preferring to present all graphics with same manner. Figure 2 differs from others.
Is it possible to add histological staining for type into proliferative (class III-IV) and non-proliferative (I-II-V) classes?
Page 7
Authors do not describe the schedule for follow-up of patients with lupus nephritis in section materials and methods. Unfortunately, it is not clearly, from section "3.4. Evolution of Clinical and Immunological Parameters" if all 401 patients were included in for follow-up schedule. Moreover, duration between controls 1-3 must be written.
Page 4 line 112 and Page 12, line 206
The abbreviation for platelets is PLT, not Plq.
Page 14, line 288
Authors indicate that this study was approved by the ethics committee of the Clínica de la Costa. Unfortunately, the protocol number is not presented in this text. Please add protocol number of ethics committee.
Conclusions are too short.
Author Response
- The bibliographic references were reviewed and adjusted.
- The abbreviation "NL" was utilized.
- The exclusion criteria applicable to the subjects were detailed in the "Subjects" section.
- In Figure 1, the duplication of the p-value was removed.
- The p-value in Figure 2 was a typographical error and was corrected.
- The color scheme of the graphs was standardized.
- Histopathology was presented as Non-proliferative (I, II, V) and Proliferative (III, IV).
- Unfortunately, the temporality could not be established nor unified, as the Colombian healthcare system does not define specific dates for patient care, and the time interval between follow-ups can vary between 3 and 9 months.
- The abbreviation for platelets was adjusted to "Plq."
- The date of protocol approval by the committee and the approval sequence from Universidad Simón Bolívar were included.
Reviewer 2 Report
Comments and Suggestions for Authors
The paper titled: Clinical and Immunological Factors Associated with the Progression of Lupus Nephritis in a Population from the Colombian Caribbean (ID biomedicines-2950199) represents an exciting analysis of 401 patients with Lupus nephritis during SLE. Authors stated that the initial proteinuria >2 g/day increases the risk factor of lack of therapy response, while mortality is associated with serum creatinine >1.5 mg/dl (p=0.01) and thrombocytopenia (p=0.01).
1. The units should be changed in Table 2: the Proteinuria (g/día), and in Table 6 - PRT >2 g/día (g/day?)
2. The inclusion criteria of SLE required the presence of ANA. The Authors described only anti-ds-DNA. What about other antibodies typical for SLE, e.g., anti-SM or antiphospholipid antibodies? Were they present and correlated with disease activity or kidney involvement?
3. Authors wrote: Mycophenolate Mofetil was the most 238
used immunosuppressant in 70% of cases. However, 52% of evaluated patients showed 239
no response to treatment, with only 28% achieving total response, consistent with other 240
reports where the total response was around 20.2%, possibly reflecting the severity of the dis- 241 ease [14].
What was the dose of Mycophenolate Mofetil? Was it the same in all patients?
4. Did you find a correlation between the dose of Mycophenolate Mofetil and the disease activity?
5. How will you treat the patient with the same histopathological changes (e.g., class IV), creatinine 2mg/dl, but different platelet levels (thrombocytopenia vs. average PLT count)? The indications of treatment based on your data can be valuable from a clinical point of view.
Author Response
- Units adjusted.
- It is true that the criteria for SLE include ANA. These patients were already diagnosed with SLE and had consequent NL. They were not evaluated regarding SLE because it was not the focus of the study, and all patients had renal involvement. However, this is an important note for future studies.
- Previous studies have reported atypical behavior in the Caribbean Region, and this severity may be associated with late presentation to the healthcare system.
- The standard dose was 2-3 g/day and was specified in the document.
- We cannot state this with certainty, as this aspect was not evaluated. The dose was adjusted according to the patient's tolerance and needs.
- We understand the value of the presented information and are very grateful for it. However, clinical indications should be balanced in clinical studies, and the management of each patient should be carefully adapted considering hematological status and the severity of renal disease. The objective of this study was to highlight the behavior of our patients, not to conduct a clinical response evaluation study.
Reviewer 3 Report
Comments and Suggestions for Authors
A paper by Vélez-Verbel et al. describes clinical and immunological factors related to the progression of lupus nephritis (LN) in a population from the Colombian Caribbean. The paper presents important findings, however, I have several comments.
Please use a short form for lupus nephritis, LN, along with the manuscript.
Please recheck the abstract according to the MDPI suggestions.
Line 48, Black population? Please recheck it.
Lines 50-51, please be more specific when describing such factors.
Lines 62-63. Please summarize the knowledge of these factors and cite relevant papers.
The introduction section lacks of epidemiology, clinical course and diagnostic procedures in LN. Moreover, it should be more catchy for readers, hence, it is one of the most important parts of the manuscript. I believe rewriting the whole section should be applied.
Line 67. Were all LN patients diagnosed with renal biopsy?
Line 79. Why did you use media with IRQ? Non-normality of the data? Should be Q1-Q3 ranges, please change it also in the text, e.g. line 95.
What about the class No. VI?
Were LN patients treated with glucocorticosteroids? What about other immunosuppressive treatments?
Did you collect data on BMI?
Line 104. Q1-Q3 ranges actually.
What were the timelines in which you assessed deaths in LN patients?
Cyclophosphamide (CTX).
Table 2. Please calculate p-values for all comparisons.
What about white blood cells?
Anti-dsDNA. Presence of titer? Please recalculate and add Q1-Q3 ranges.
Gender should be replaced with sex in the whole manuscript.
In case of very small p-values, please write p<0.001.
Figure 1 presents something that is written in the text, with even exact values. What is the sense of doubling it?
Lines 123-125. Which test did you use to analyze the data? Please write p = 0.033 in the Figure 2. What does this p-value stand for?
Platelets, better 103/μl (tables). Line 171 also, etc.
Tables 2. Please provide exact p-values.
What was the timeline of controls in LN patients?
Sex, female vs. male.
How would you interpret an interesting finding in Table 5 regarding anti-dsDNA between deceased and survivors?
Line 31. Please write 95% odds ratio also like in the Results section.
Lines 208-228. This part should be in the Discussion section with in-depth analysis based on the literature review, more interpretation of the data is needed (!).
Line 230, ref.?
The concept of looking for factors in LN and its prognosis is interesting, nevertheless, nowadays we need to know more about the disease course, thus, elaborating specific comparisons in the group of LN might be useful. Next, recent papers on the LN evaluating its onset, e.g. early vs. delayed, could add some novel information in the Discussion section (doi: 10.1007/s00296-024-05579-4).
Line 274. I am not sure if based on your results you can speculate about the whole Colombian Caribbean population.
Please provide a number of the approval from the Bioethics Committee in the Methods section.
Please check the reference list according to the MDPI regulations.
Comments on the Quality of English LanguageMinor editing of English language required.
Author Response
### Manuscript Adjustments and Translation for Clinical Journal
- Please use the abbreviated form for lupus nephritis, LN, throughout the manuscript.
- Response: Adjusted
- Please revise the abstract according to the MDPI suggestions.
- Response: Adjusted
- Line 48, Black population? Please recheck this.
- Response: Changed to Afro-descendant
- Lines 50-51, please be more specific when describing these factors.
- Response: Described in more detail
- Lines 62-63. Please summarize the knowledge of these factors and cite relevant papers.
- Response: Completed
- The introduction section lacks information on the epidemiology, clinical course, and diagnostic procedures in LN. Additionally, it should be more engaging for readers, as it is one of the most important parts of the manuscript. I believe the entire section should be rewritten.
- Response: Adjusted
- Line 67. Were all LN patients diagnosed with renal biopsy?
- Response: All patients had renal biopsy
- Line 79. Why did you use median with IQR? Non-normality of the data? It should be Q1-Q3 ranges; please change it also in the text, e.g., line 95.
- Response: The use of statistical measures depends on the normality of the data. A Kolmogorov-Smirnov test was performed to define the presentation. IQR is interquartile range and is defined as IQR = Q3 - Q1. It can be presented as Median (Q₁, Q₃) in the text, specifying Q₁ and Q₃.
- What about class No. VI?
- Response: Not considered since these patients are on dialysis and/or transplanted, representing a subgroup with specific management needs and different prognosis that could affect the results.
- Were LN patients treated with glucocorticoids? What about other immunosuppressive treatments?
- Response: Patients were treated according to the standard protocol, which includes pulses of methylprednisolone and subsequent maintenance at low doses.
- Did you collect data on BMI?
- Response: Not considered for the analysis
- Line 104. Actually, Q1-Q3 ranges.
- Response: Adjusted
- What were the timelines in which you assessed deaths in LN patients?
- Response: Each time the national health system is updated, every three or four months.
- Cyclophosphamide (CTX).
- Response: Adjusted
- Table 2. Please calculate p-values for all comparisons.
- Response: Completed
- What about white blood cells?
- Response: Included
- Anti-dsDNA. Presence of titer? Please recalculate and add Q1-Q3 ranges.
- Response: Reported as Positive/Negative, in % positivity, and the graph was made in IU. The cutoff points for positivity/negativity were established by:
Gargiulo MA, Khoury M, Gómez G, Grimaudo S, Suárez L, Collado MV, Sarano J. Cut-off values of immunological tests to identify patients at high risk of severe lupus nephritis. Medicina (B Aires). 2018;78(5):329-335. Available at: https://pubmed.ncbi.nlm.nih.gov/30285925/
- Gender should be replaced with sex throughout the manuscript.
- Response: Completed
- In case of very small p-values, please write p<0.001.
- Response: Completed
- Figure 1 presents something that is written in the text, with even exact values. What is the sense of doubling it?
- Response: Removed
- Lines 123-125. Which test did you use to analyze the data? Please write p = 0.033 in Figure 2. What does this p-value stand for?
- Response: The statistical tests used are mentioned between lines 90 and 94, and each table footnote contains a superscript indicating the specific test used. The p-value in Figure 2 was removed; it was a transcription error.
- Platelets, better 10³/μl (tables). Also in line 171, etc.
- Response: Adjusted
- Tables 2. Please provide exact p-values.
- Response: Provided
- What was the timeline of controls in LN patients?
- Response: Unfortunately, temporality could not be established or unified, as the Colombian health system does not define specific dates for patient care. The time interval between controls can vary between 3 and 9 months.
- Sex, female vs. male.
- Response: Adjusted
- How would you interpret an interesting finding in Table 5 regarding anti-dsDNA between deceased and survivors?
- Response: Higher positivity is associated with higher mortality.
- Line 31. Please write the 95% odds ratio also like in the Results section.
- Response: Completed
- Lines 208-228. This part should be in the Discussion section with in-depth analysis based on the literature review; more interpretation of the data is needed (!).
- Response: This is a result and corresponds to the results section, but it has been expanded in the discussion section.
- Line 230, ref.?
- Response: Provided
- The concept of looking for factors in LN and its prognosis is interesting; however, nowadays we need to know more about the disease course, thus, elaborating specific comparisons in the LN group might be useful. Next, recent papers on LN evaluating its onset, e.g., early vs. delayed, could add some novel information in the Discussion section (doi: 10.1007/s00296-024-05579-4).
- Response: Information added
- Line 274. I am not sure if based on your results you can speculate about the whole Colombian Caribbean population.
- Response: More than speculation, this is an inherent reality of our region. This study is part of various ongoing research projects in the Colombian Caribbean region within our research group, as the center where this study was conducted is a regional reference center, being the only one that provides renal biopsy, nephropathology, and follow-up program services.
- Please provide an approval number from the Bioethics Committee in the Methods section.
- Response: Provided
- Please check the reference list according to MDPI regulations.
- Response: Checked and adjusted
Round 2
Reviewer 3 Report
Comments and Suggestions for Authors
The paper has been partially improved. Needs more revisions.
Line 88. as medians and quartile one and three ranges (Q₁-Q₃). => as medians with Q1-Q3 ranges.
(1) => [1]
Line 93. Chi2
Table 1. Class VI 0 (0.0%)
Tables. Should be rounded the the second decimal place. If p<0.05, to the third, or p<0.001.
Figure 2. Gender => Sex
Please see the discussion.
Comments on the Quality of English LanguageMinor editing of English language required.
Author Response
- Line 88. as medians and quartile one and three ranges (Q₁-Q₃). => as medians with Q1-Q3 ranges.=> [1]
Done
- Line 93. Chi2
Done
- Table 1. Class VI 0 (0.0%)
Done
- Should be rounded the the second decimal place. If p<0.05, to the third, or p<0.001.
Done
- Figure 2. Gender => Sex
Done
In addition it is essential that they:
* Expand the discussion further exploring the wider literature in the field. Does there data alude to any previousl published literature that might provide mechanistic insight to these observations or not.
The discussion was expanded and new references were included regarding the topic.
* Abstract: authors should better define non responders
Done
* Final Grammar and English language proofing is required as I identified several instances of poor sentence structure.
A review of the English language was conducted, and the necessary adjustments were made.